# Does Data Availability Constrain Temperature-Index Snow Models? A Case Study in a Humid Boreal Forest

**Achut Parajuli** [1,2,*] **, Daniel F. Nadeau** [1,2] **, François Anctil** [1,2] **, Oliver S. Schilling** [1,2,3] **and Sylvain Jutras** [2,4]

1   Department of Civil and Water Engineering, Université Laval, Quebec City, QC G1V 0A6, Canada; daniel.nadeau@gci.ulaval.ca (D.F.N.); francois.anctil@gci.ulaval.ca (F.A.); oliver.schilling@unine.ch (O.S.S.)
2   CentrEau, Québec Water Research Centre, Université Laval, Quebec City, QC G1V 0A6, Canada; sylvain.jutras@sbf.ulaval.ca
3   Centre for Hydrogeology and Geothermics, Université de Neuchâtel, 2000 Neuchâtel, Switzerland
4   Department of Wood and Forest Science, Université Laval, Quebec City, QC G1V 0A6, Canada
*   Correspondence: achut.parajuli.1@ulaval.ca

**Abstract:** Temperature-index (TI) models are commonly used to simulate the volume and occurrence of meltwater in snow-fed catchments. TI models have varying levels of complexity but are all based on air temperature observations. The quality and availability of data that drive these models affect their predictive ability, particularly given that they are frequently applied in remote environments. This study investigates the performance of non-calibrated TI models in simulating the subcanopy snow water equivalent (SWE) of a small watershed located in Eastern Canada, for which some distinctive observations were collected. Among three relatively simple TI algorithms, the model that performed the best was selected based on the average percent bias (*Pbias* of 24%) and root mean square error (*RMSE* of 100 mm w.e.), and was designated as the base TI model. Then, a series of supplemental tests were conducted in order to quantify the performance gain that resulted from including the following inputs/processes to the base TI model: subcanopy incoming radiation, canopy interception, snow surface temperature, sublimation, and cold content. As a final test, all the above modifications were performed simultaneously. Our results reveal that, with the exception of snow sublimation (*Pbias* of 5.4%) and snow surface temperature, the variables mentioned above were unable to improve TI models within our sites. It is therefore worth exploring other feasible alternatives to existing TI models in complex forested environments.

**Keywords:** snowmelt; temperature-index model; boreal forest; data quality; data availability

## 1. Introduction

Of all the components in the cryosphere, snow covers the greatest area in wintertime [1]. Snowmelt provides more than 50% of the annual runoff for the majority of catchments in the Northern Hemisphere [2]. Therefore, snowpack dynamics heavily affect streamflow patterns [3], making snow accumulation and melt crucial aspects to consider for accurate hydrological simulations and forecasting [4]. Approximately 19% of the snow cover within the Northern Hemisphere is located in forested areas [5,6], where snow accumulation and melting are different than in open environments.

There have been substantial efforts to develop snowmelt models [3,7], from simple empirical approaches [8–11] to more complex ones that include the influence of forest canopies [3,12–17]. Regardless of the snow model, there have been persistent weaknesses in data quality and availability in the modelling chain [18]. For instance, all snow models require precipitation input, but it has been well established that accurate measurements of solid precipitation are hindered by wind-induced

undercatch, which may reach 20% to 70% depending on the gauge type, shielding method and prevailing meteorological conditions [19,20]. Precipitation observations therefore require adjustments whenever accurate snow accumulation and ablation estimates are needed [21]. Similarly, the inaccurate separation of precipitation phases (solid/liquid) may lead to undesired biases when modelling the snow water equivalent (SWE) [22–24]. Uncertainty associated with sublimation and blowing snow transport, both of which are difficult to quantify, are other aspects that equally affect melt modelling [21].

Snow models can be organized into two main categories: energy balance (EB) models [12–14,16,25–27] and temperature-index (TI) models [3,28–36]. They all aim to estimate how the snow water equivalent (SWE), which ultimately affects streamflow dynamics, evolves throughout winter.

In EB models, melting is inferred from the balance of heat flux exchanged between the surface of the snowpack and the atmosphere, as well as between the base of the snowpack and the ground [37–39]. Such models have been adapted for mountainous forested areas by using an explicit representation of canopy interception and sublimation losses [12–14]. However, for operational applications, these models require more input variables (wind speed, incoming solar radiation) than routinely available [31]. Besides, difficulties in the spatial interpolation of variables, such as wind speed and incoming radiative fluxes, hinder their applicability on larger scales [3], especially when requiring sophisticated and expensive sensors. For research applications, reanalysis datasets such as ERA5 provide an option where these inputs are readily available at hourly time steps [39]. Nevertheless, one of the shortcomings of such a dataset is its resolution (i.e., 31 km × 31 km), thus creating challenges when undertaking small-scale studies. A number of hydrological applications require a more simplistic approach where only the relevant processes are thoroughly addressed [40,41]. As such, several studies argue in favour of simpler and widely used TI models, which remain a popular choice to simulate snow and ice melt for catchment scale hydrological applications [3,11,29,31,41–47].

The simplest version of a TI model employs a unique parameter known as the degree-day (or melt) factor, and uses air temperature as a proxy for the energy available above the snow cover [3,29]. According to Ohmura [48], air temperature acts as a controlling variable for the sensible heat flux and incoming longwave radiation, which is also correlated with shortwave radiation. These three energy components are important drivers of snow and ice melt [31]. However, TI models that rely solely on air temperature are often unable to fully describe the spatiotemporal variability in snowmelt rates [49]. In particular, these models are unable to incorporate the influence of spatially variable solar radiation, diurnal fluctuations, and forest canopies. As such, there has been a series of advancements proposed over the last few years to incorporate more explicit snow processes into TI based models [3,29,48]. For instance, Hock [29] added shortwave solar radiation over a regularly spaced grid (20 m resolution) in order to account for sub-daily snow melt. More recently, Tobin et al. [31] proposed a time variable melt factor following a sinusoidal function to capture sub-daily melt with improved accuracy. Cold content was also incorporated into some TI models to better simulate snow [32,33] and glacier melt [50], as was done by Jost et al. [3], who focused on forested environments. Although snow surface temperature and sublimation are relevant inputs, these data are rarely available due to difficulties associated with their monitoring in the field, and therefore several previous studies have resorted to TI models that assume a fixed threshold temperature [3,29,31,41].

In order to identify key processes to be included in forest snow modelling, the availability of observations describing all snow cover conditions remains an important limitation. This issue must be further explored in terms of volume (snow depth and water equivalent) and thermal regime (air, snow surface, and snowpack temperature). Parajuli et al. [51] explored the spatial and temporal variability of the snow water equivalent (SWE) in a small boreal forest watershed. Building on that work, this study takes advantage of an experimental site that combines very detailed observations of typical TI model inputs (bias-corrected solid precipitation, distributed air temperature measurements) along with other potential inputs such as snow temperature, snow height, SWE, and turbulent and radiative fluxes between the canopy and the atmosphere. The availability of such a dataset allows us to articulate

the following research question: Does data availability constrain SWE modelling based on the TI approach in a humid boreal forest catchment? Three variants of the TI algorithm were first tested on detailed observations collected during two winters. The best performing algorithm was then subjected to additional testing using subcanopy incoming radiation, sublimation, cold content, snow surface temperature, and canopy interception. By conducting a series of tests that include one process/input at a time and quantifying the associated performance, we seek to highlight the relevance of taking these processes into account in a TI model.

## 2. Material and Methods

### 2.1. Study Area

This study was conducted in the experimental watershed "*Bassin Expérimental du Ruisseau des Eaux-Volées*" (BEREV) in Montmorency Forest in Quebec, Canada. It focused on sub-basin 7 that drains an area of 3.49 km² (Figure 1), with an average slope of 10.7° and elevation ranging from 725 to 977 m ASL [51]. The region is characterized by a cold and humid continental climate with 1583 mm of mean annual precipitation [52], out of which 40% falls in solid form. The seasonal snow cover normally starts in early November and persists until late May. This region lies in the Laurentian Mountains in which the U-shaped valley is covered with glacial deposits [51]. The long-term (1981–2010) average winter (November to May) temperature at and around the site is −6 °C [53]. The local pedoclimatic conditions favour the growth of balsam fir (*Abies balsamea*), with some occurrences of white birch (*Betula papyrifera*), white spruce (*Picea glauca* (Moench.)), and black spruce (*Picea mariana* (Mill.)) [54]. Due to past logging operations, there are several patches of forest clearings accompanied by spatially varying stand structures (Figure 1). For detailed descriptions of the study area, please refer to Parajuli et al. [51], Isabelle et al. [52,55] and Hadiwijaya et al. [54].

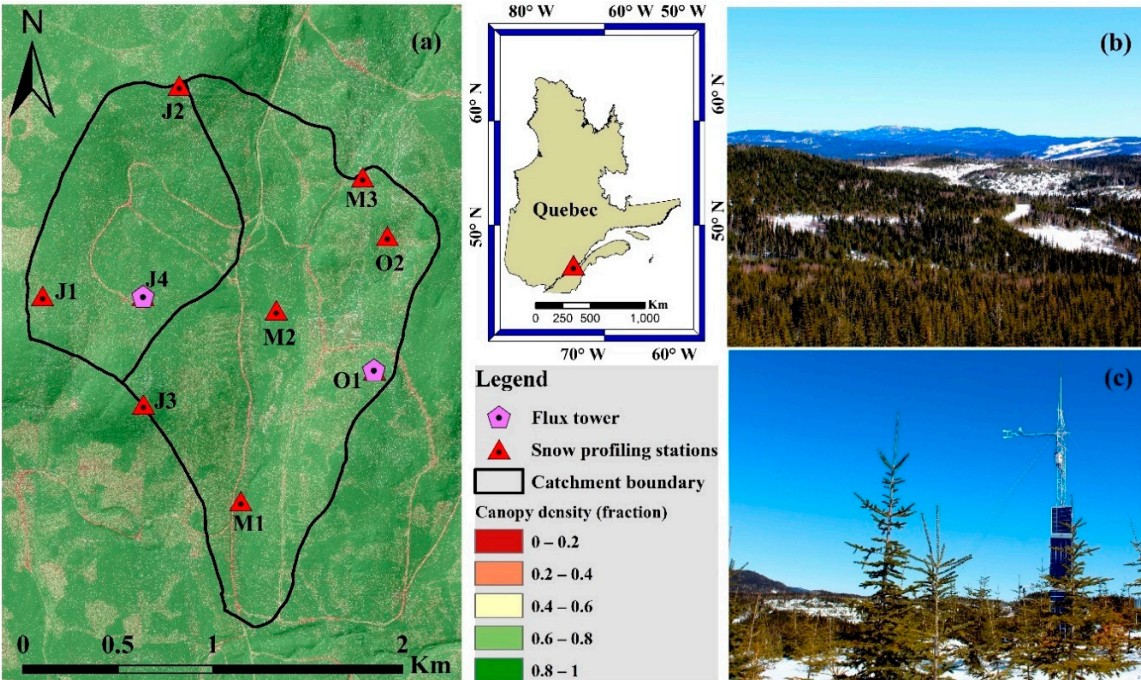

**Figure 1.** Study area. (**a**) Map of basin 7 within the "*Bassin Expérimental du Ruisseau des Eaux-Volées*" (BEREV), illustrating the locations of the snow-profiling stations where extensive snow water equivalent (SWE) sampling was performed during winters 2016–2017 (W1) and 2017–2018 (W2). Top right of the map: the location of BEREV in Quebec, Canada. (**b**) Flux tower near site O2. (**c**) Typical winter conditions at the BEREV.

## 2.2. Experimental Setup

The experimental design is based on nine study sites spread across the watershed, representing a diversity of features in terms of topography and canopy structure (Table 1). Of these sites, two are open areas (mean tree height < 3 m), while the others have forest cover (mean tree height > 3 m). Each site is equipped with a custom-made snow-profiling pylon with 18 T-type thermocouples and a snow depth sensor (see Section 2.3.5) that measure the evolution of the seasonal snowpack. An additional T-type thermocouple is enclosed within a radiation shield located 2 m above ground in order to extract a time series of the near-surface air temperature.

**Table 1.** General forest characteristics of the study sites, where LAI is the leaf area index and Csc is the canopy density (1 is a closed canopy and 0 is an open site).

| Site | LAI ($m^2$ $m^{-2}$) | Csc (−) | Aspect (°) | Tree Height (m) | Forest Cover |
|------|------|------|------|------|------|
| O1 | 0 | 0 | 160 | 0 | forest clearing |
| O2 | 2.58 | 0.86 | 277 | 1.8 | sapling |
| J1 | 2.89 | 0.63 | 33 | 3.1 | juvenile |
| J2 | 3.80 | 0.86 | 351 | 4.3 | juvenile |
| J3 | 3.61 | 0.88 | 96 | 5.3 | juvenile |
| J4 | 3.40 | 0.76 | 334 | 8.1 | juvenile |
| M1 | 3.52 | 0.97 | 124 | 8.6 | mature |
| M2 | 2.29 | 0.71 | 220 | 12.5 | mature |
| M3 | 2.09 | 0.95 | 320 | 13.3 | mature |

Two flux towers (Figure 1) are also in operation in basin 7, and are located in the vicinity of snow-profiling stations J4 and O2. They are equipped with eddy covariance sensors (IRGASON, Campbell Scientific, UT, USA) that measure turbulent fluxes of heat and water vapour (snow sublimation and evapotranspiration), and CNR4 net radiometers (Kipp and Zonen, Delft, the Netherlands). Further details about the flux towers is provided by Isabelle et al. [55]. An intensive manual snow survey was conducted during winters 2016–2017 (W1) and 2017–2018 (W2) around each snow-profiling station, for a total of 1061 observations. These samples were collected from a subplot nearby the snow profiling station, where a minimum of 5 and a maximum of 12 samples per visit were collected depending on the exact canopy structure. More details on the sampling design, size, frequency, and limitations of these field measurements can be found in Parajuli et al. [51].

## 2.3. Data Inputs for TI Model

To run a simple TI model, one obviously requires air temperature and precipitation time series. In this study, all other input variables tested are also described below. All the data inputs required to run TI models were aggregated to an hourly timestep.

### 2.3.1. Air Temperature

Air temperature data were extracted locally at each snow profiling station. Since the series of air temperature observations at site J4 was nearly complete, these measurements were used to fill in the missing data at the other sites, accounting for the mean vertical temperature gradient between the two sites.

### 2.3.2. Precipitation

Precipitation data were obtained from the federal weather station (ECCC ID: 7042388) [53] located 4 km north of the research site. Precipitation was assumed to be uniformly distributed throughout the (small) basin. To adjust for wind-induced undercatch bias in the precipitation data, we used manually (twice daily) recorded precipitation (nearby the federal weather station) from a Double Fence Intercomparison Reference (DFIR) gauge and applied a simple adjustment by

calculating the bias between the DFIR and Environment and Climate Change Canada (ECCC) extracted precipitation. We then corrected the undercatch by adding bias fractions to the hourly precipitation input. Following Jennings et al. [22], precipitation was assumed to be falling in its solid state when air temperatures were below 0 °C, liquid when air temperatures were above 2 °C and linearly distributed between the two phases for air temperatures between 0 and 2 °C.

### 2.3.3. Sublimation

Sublimation data were obtained from raw turbulence data measured at 10 Hz using the data processing software Eddy Pro $^{©}$, version 6.0 (LI-COR Biosciences, NE, USA). In short, this software makes it possible to eliminate data outliers and to apply various corrections in relation to the coordinate system, density effects, etc. These data were available at 30-min spans and averaged at hourly timesteps similar to the other model inputs. Details on the treatment of the eddy covariance data are provided by Isabelle et al. [55].

### 2.3.4. Snowpack and Snow Surface Temperature

Snow surface and snowpack temperatures were measured at all profiling stations. The thermocouples were vertically installed every 10 cm from the ground-level to a height of 1.8 m, providing a vertical profile of snowpack temperatures. Snow surface temperatures were estimated by taking the measurements reported by the thermocouple in the snowpack closest to the surface. Most of the data gaps in snow surface and snowpack temperatures were concentrated during the melting period. Based on our field-based snow pit surveys, the snowpack temperature during that period typically ranged between −0.2 and −0.7 °C; thus, existing gaps were replaced by a constant temperature of −0.45 °C. To fill the snow surface temperature gaps at our sites, we utilized the dataset recorded by the snow profiling stations that measured the snowpack temperatures closest to the snow surface.

### 2.3.5. Snow Depth

An ultrasonic snow depth sensor (Judd Communication, Salt Lake City, UT, USA) was mounted at each site, except for site O2, where a SR50 (Campbell Scientific, Salt Lake City, UT, USA) was mounted instead. These sensors provided the time series of snow depth at an hourly time step.

### *2.4. Complementary Measurements*

On each forested site (sites J1 to J4 and M1 to M3 in Table 1), the leaf area index (LAI) was determined during summer 2017 using a hemispherical camera system (WinSCANOPY, Regent Instruments, QC, Canada.) Additionally, vegetation (tree height, and canopy density) and topographic information (digital elevation model) were derived from LiDAR data (1-m resolution) collected in 2016.

### *2.5. Melt Modelling*

Figure 2 summarizes the modelling strategy, which is divided into two phases. The first one aims to determine the best model or base model, and the second one aims to evaluate the performance gains associated with the addition of supplemental processes.

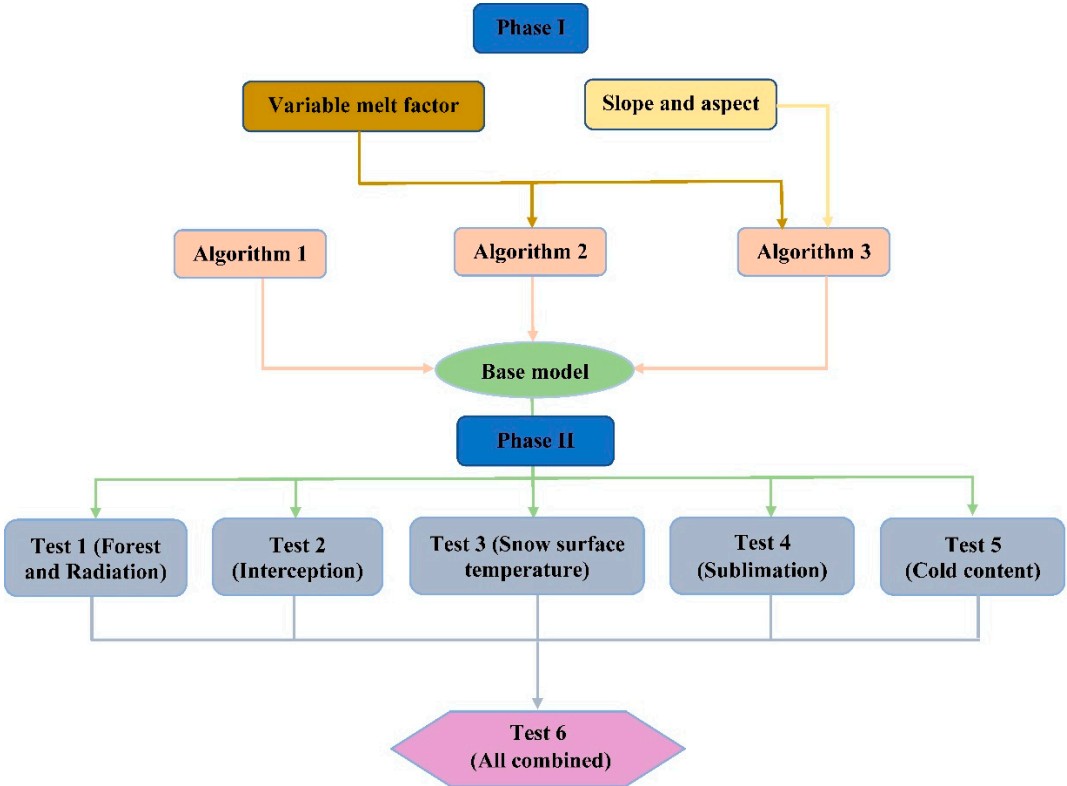

**Figure 2.** Flow diagram illustrating the experimental design of this study.

### 2.5.1. Candidates for the Base TI Model (Phase I)

During our search for the base TI model, we deliberately neglected key components such as snow interception and solar radiation and other relevant inputs/processes that affect snow accumulation and its melt. These variables were included in the second phase of testing (Section 2.5.2) to isolate their associated gains. The three potential candidates for our base TI model is given below:

Algorithm 1 (A1)

As a first test candidate of our base TI model, we employed the simplest version of a TI model (A1):

$$M = \begin{cases} m_{ft} \, (T_a - T_t) & T_{air} > 0 \, °C \\ 0 & T_{air} \leq 0 \, °C \end{cases} \tag{1}$$

where $M$ (mm h$^{-1}$) is the potential snow and ice melt at an hourly time step, $m_{ft}$ (mm °C$^{-1}$ day$^{-1}$) is the base melt factor, $T_a$ (°C) is an hourly air temperature series measured 2 m above the ground by the snow profiling station, and $T_t$ is the threshold temperature (taken as 0 °C), usually considered as a constant in the range of 0–2 °C [29,31,41]. For uniformity, we divided the daily base melt factor ($m_{ft}$) by 24 (i.e., mm °C$^{-1}$ day$^{-1}$ to mm °C$^{-1}$ h$^{-1}$).

Algorithm 2 (A2)

As the second candidate for the base TI model (A2), Equation (1) was used to derive the potential snowmelt $M$, and the melt factor $m_f$ is substituted by the expression presented in Tobin et al. [31]:

$$m_{td} = \begin{cases} m_f + \beta \Delta_T \sin \left( \pi \frac{t_d - t_0}{t_1 - t_0} \right) & t_0 \leq t_d < t_1 \\ m_f + \beta \Delta_T Z & \text{otherwise} \end{cases} \tag{2}$$

where $m_{td}$ (mm °C$^{-1}$ day$^{-1}$) is the time variable melt factor, $t_0$ (h) is the start of daylight, $t_d$ (h) is the current timestep of the day, $t_1$ (h) is the end of daylight on the current day, $\Delta_T$ (°C) is the difference between maximum and minimum temperatures, which is derived from air temperature ($T_a$) collected by the snow profiling station at 2 m above the ground, and $\beta$ is a factor to convert temperature amplitude into the melt factor (mm °C$^{-1}$ day$^{-1}$). $Z$ is the factor ensuring the daily mean value of $m_{td}$ is equal to $m_f$, depending on the length of the day $l_n$ (h):

$$Z = 2\left(\frac{t_1 - t_0}{\pi l_n}\right) \tag{3}$$

Algorithm 3 (A3)

The third candidate we explored was a snow module implemented in the HBV (Hydrologiska Byråns Vattenbalansavdelning) model [56], which is a popular approach in hydrological studies [3]. Hamilton et al. [57] modified this module by adding slope and aspect components, as presented in Jost et al. [3]:

$$M = \begin{cases} m_{td}m_f\,(1 - (b_{hbv}\sin(s)\cos(a)))(T_a - T_t) & T_a > 0\,°\mathrm{C} \\ 0 & T_a < 0\,°\mathrm{C} \end{cases} \tag{4}$$

where, $a$ is the aspect (°), $s$ is the slope (°), and $b_{hbv}$ is the conversion factor to change ° (slope/aspect) to mm °C$^{-1}$ day$^{-1}$. Here, $m_{td}$ is defined by Equation (4).

### 2.5.2. Impacts of Including Additional Model Processes/Inputs (Phase II)

Once the base TI model was identified by comparing it with local observations, we then evaluated the impact of including additional processes in the SWE modelling (Figure 1, phase II).

### 2.5.2.1. Test 1: Subcanopy Incoming Radiation

The first model we tested incorporated solar radiation and canopy effects following Jost et al. [3]:

$$M = \begin{cases} m_f\,(T_a - T_t) + (r_f K_n) & T_{air} > 0\,°\mathrm{C} \\ 0 & T_{air} < 0\,°\mathrm{C} \end{cases} \tag{5}$$

where $r_f$ (mm m$^2$ W$^{-1}$ h$^{-1}$) is the radiation factor, $K_n$ (W m$^{-2}$) is the attenuated solar radiation beneath the canopy, which is derived from Allen et al. [58] following the Beer–Lambert extinction law as described by Isabelle et al. [52], and $m_f$ (mm °C$^{-1}$ h$^{-1}$) is the modified melt factor to account for canopy-related effects, defined as:

$$m_f = m_{ft}(1 - C_{cs}) + b_f C_{sc} \tag{6}$$

where $m_{ft}$ (mm °C$^{-1}$ h$^{-1}$) is the base melt factor, $b_f$ is the melt factor for forested areas, and $C_{sc}$ is the canopy density (fraction).

### 2.5.2.2. Test 2: Canopy Interception

For the second test, we added the snow interception process to the base TI model. Canopy interception is modelled using the approach presented in Barlett et al. [59] and Hedstrom and Pomeroy [60]:

$$I = I_1 exp^{-Ut}, \; I_1 = (I^* - L_0)\left(1 - exp^{C_{cs}\frac{P}{I^*}}\right) \tag{7}$$

where $I$ (mm) is the snow interception, $I_1$ (mm) is the snow interception that occurs before the unloading phenomenon, $I^*$ (mm) is the maximum canopy load, $L_0$ (mm) is the initial intercepted snow load from the end of the previous time step, $P$ (mm) is the snowfall amount, $U$ is the unloading coefficient, and $t$

(h) is the time since the previous snowfall event. As described in Hedstrom and Pomeroy [60], we used a method developed by Schmidt and Gluns [61] to estimate the maximum snow load:

$$I^* = \overline{S}\left(0.27 + \frac{46}{\rho_{sf}}\right)LAI \tag{8}$$

where, $\overline{S}$ (kg m$^{-2}$) is the species coefficient (using 6 kg m$^{-2}$ [59] as a reference value for boreal tree species), $\rho_{sf}$ (kg m$^{-3}$) is the fresh snow density, which is derived using air temperature ($T_a$) and Equation (3), given as:

$$\rho_{sf} = 67.92 + 51.25\, e^{\left(\frac{T_a}{2.59}\right)} \tag{9}$$

In the above interception model, Hedstrom and Pomeroy [60] were unable to estimate the value of $U$ and suggested $exp^{-Ut}$ to be 0.678 based on weekly tests conducted over four years. The same value is used here to derive snow unloading.

### 2.5.2.3. Test 3: Variable Threshold Temperature

As mentioned above, TI models commonly use threshold temperatures in the range of 0 to 2 °C. Thus, our third test was designed to assess the impact of providing the measured surface temperature of the snow cover instead of using a constant threshold.

### 2.5.2.4. Test 4: Sublimation

Snow sublimation is difficult to monitor in situ as it requires sophisticated instrumentation [62]. As such, TI models are typically supplied with ad hoc average sublimation values distributed equally among the studied period (e.g., 3). Supported by eddy covariance sensors, this study aims to identify the effect of inserting direct observation of snow sublimation into the base TI model. To do so, two avenues are explored. In the first approach, we supplied the TI model with sublimation data recorded above a juvenile forest (site J4) for sites with a canopy, while in our second approach, we supplied the model with sublimation data that were recorded at the sapling site (near O2) for sites with shorter vegetation.

### 2.5.2.5. Test 5: Cold Content

The snowpack cold content is given by:

$$CC = c_i \rho_{sa} D_s \left(T_{sp} - T_0\right) \tag{10}$$

where $c_i$ (MJ kg$^{-1}$ °C$^{-1}$) is the specific heat of ice, $\rho_{sa}$ (kg m$^{-3}$) is the snow density, $D_s$ (m) is the snow depth, and $T_{sp}$ (°C) is the snowpack temperature. Snow density undergoes gravitational settlement and metamorphism over time [59]. The formulation presented in Equation (9) estimates *fresh* snow density [60,63] with no inclusion of abovementioned processes. In order to estimate the aged snow density ($\rho_{sa}$) in the above equation, we utilized the empirical formulation described in Pomeroy et al. [64], which is given as:

$$\rho_{sa} = 450 + \frac{20470}{D_s}\left(1 - e^{-D_s/67.3}\right) \tag{11}$$

Note there are obvious shortcomings when empirical formulas such as Equation (11) are used. Nevertheless, in many instances, they represent our "best guess" and as such, are often implemented in land surface models such as the Canadian Land Surface Scheme (CLASS) [59].

2.5.2.6. Test 6: Incorporating Subcanopy Radiation, Interception, T$_s$, Sublimation, and Cold Content

For the final test, we assessed the impacts of including all the processes listed in Sections 2.5.2.1–2.5.2.6 into our base TI model.

*2.6. Model Evaluation*

A possible approach for this study was to calibrate the parameters of the TI models and evaluate the model performance during validation periods. Due to the fact that we were able to obtain a very detailed dataset over the short span of two winters, we chose instead to use an approach that assumed little prior information about the watershed being studied and would therefore have to rely largely on parameters available in the literature. However, the model produced unrealistic early winter melt values when fitted using data available from the literature, and we were therefore forced to calibrate the radiation factor for our tests. Details are provided in Table 2.

**Table 2.** Set of parameters used in this study.

| Variable | Symbol | Value | Unit | Reference |
|---|---|---|---|---|
| base melt factor | $m_f$ | 3.74 | mm °C$^{-1}$ day$^{-1}$ | [33] |
| radiation factor | $r_f$ | $1.42 \times 10^{-3}$ | mm m$^2$ W$^{-1}$ h$^{-1}$ | calibrated |
| forest factor | $b_f$ | 0.32 | mm °C$^{-1}$ day$^{-1}$ | [3] |
| degree to melt factor (a) | $b_{hbv}$ | 1.74 | - | [3] |
| specific heat capacity | $c_i$ | $2.1 \times 10^{-3}$ | MJ kg$^{-1}$ °C$^{-1}$ | constant |
| latent heat of fusion | $c_f$ | $3.34 \times 10^6$ | J kg$^{-1}$ | constant |
| temperature to melt factor | $\beta$ | $8.5 \times 10^{-4}$ | - | [31] |

To evaluate the models' performance, we utilized two criteria: the root mean square error (*RMSE*) and the percent bias (*Pbias*). *RMSE* is a popular metric that derives the deviation between modelled (*Mod*) and observed (*Obs*) samples:

$$RMSE = \sqrt{\frac{\sum_{i=i}^{n}(Mod_i - Obs_i)^2}{n}} \tag{12}$$

where *n* denotes the number of observations. The next evaluation metric, *Pbias*, provides information about the bias present in the modelled samples when compared to observations:

$$Pbias = 100 \frac{\sum_{i=1}^{n}(Mod_i - Obs_i)}{\sum_{i=1}^{n}Obs_i} \tag{13}$$

In both evaluation metrics (*RMSE* and *Pbias*), values closer to 0 are considered to provide better estimates. Here, modelled SWE was compared with the mean SWE measured at each site.

**3. Results**

*3.1. Phase I: Identifying the Base TI Model*

Figure 3 illustrates the modelled SWE derived from each of the three candidate algorithms for the base TI model. During the first winter (W1), algorithm A3 provided better SWE estimates (mean *Pbias* of −11% and *RMSE* of 81 mm w.e.) than algorithms A1 (mean *Pbias* of −20% and *RMSE* of 86 mm w.e.) and A2 (mean *Pbias* of −20% and *RMSE* of 86 mm w.e.). Over the same period, model results were more favourable at juvenile sites (J2, J3) and sites with shorter vegetation (O1 and O2). In contrast, SWE was underestimated at all mature forest sites (M1, M2 and M3). Model results followed a similar pattern over the second winter. Indeed, the candidate algorithms provided better estimates at three juvenile sites (J1, J2 and J3), but overestimated observed SWE at one juvenile site (J4) and at the mature

sites. For the second winter, algorithm A3 was again the best one (mean *Pbias* of −30% and *RMSE* of 101 mm w.e.). However, for the same period, algorithm A1 (mean *Pbias* of −32% and *RMSE* of 114.2 mm w.e.) was slightly better than algorithm A2 (mean *Pbias* of −32% and *RMSE* of 114.4 mm w.e.).

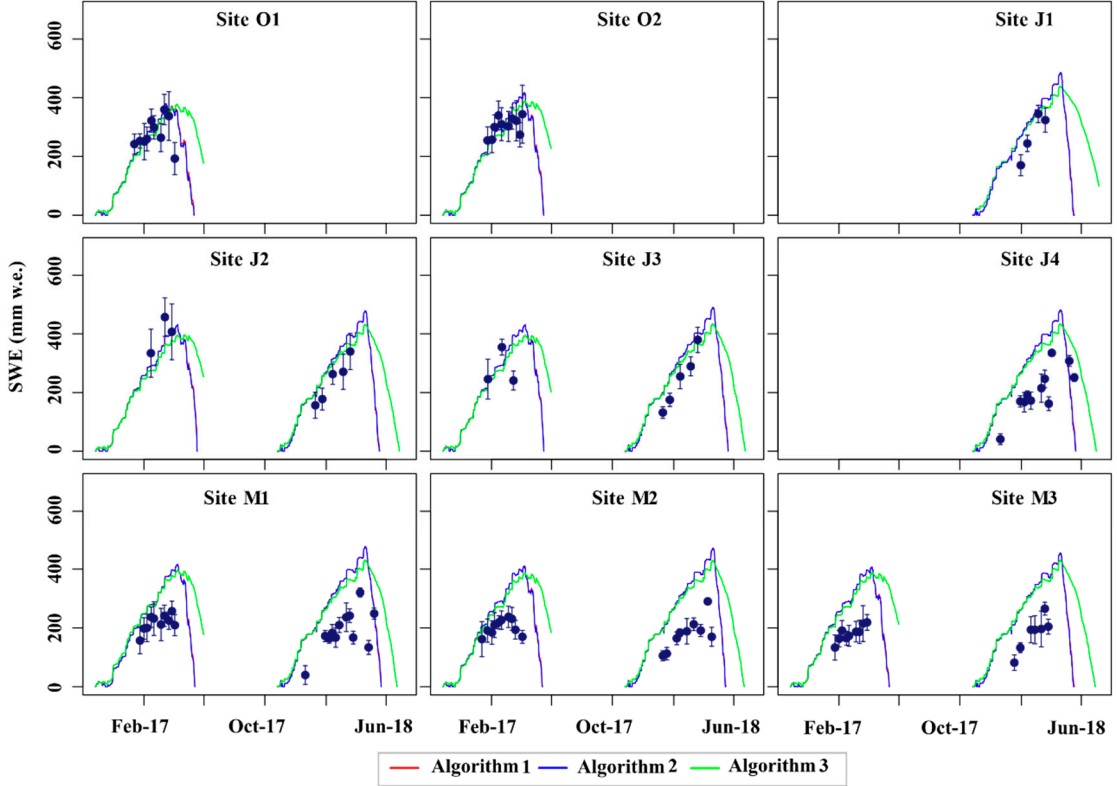

**Figure 3.** Hourly observed and simulated snow water equivalent (SWE) for the three temperature-index (TI) algorithms. Points illustrate the average SWE and the error bars denote the standard deviation. Note that the curves of algorithms A1 and A2 are almost completely superimposed.

Figure 4 illustrates algorithm performance in estimating SWE. During W1, O1 offered the best model performances, with *Pbias* values of −4%, −4% and −1% for models A1, A2 and A3, respectively. As for W2, models A1 to A3 had *Pbias* values of −19%, −19% and −16%, respectively, at site J1. Note that the abovementioned results are based on manual SWE measurements, collected mostly during the accumulation period. Based on automatic snow depth measurements and field observations, snow persisted within the catchment area until 31 May and 5 June during W1 and W2, respectively. Figure 5 compares the timing of snow disappearance simulated by all three algorithms (in days) with observations (automatic snow depth sensor). A1 (−6.9 days all site average) and A2 (−7.8 days all site average) somehow anticipated the observed disappearance, while A3 considerably lagged in that respect (+37.9 days all site average).

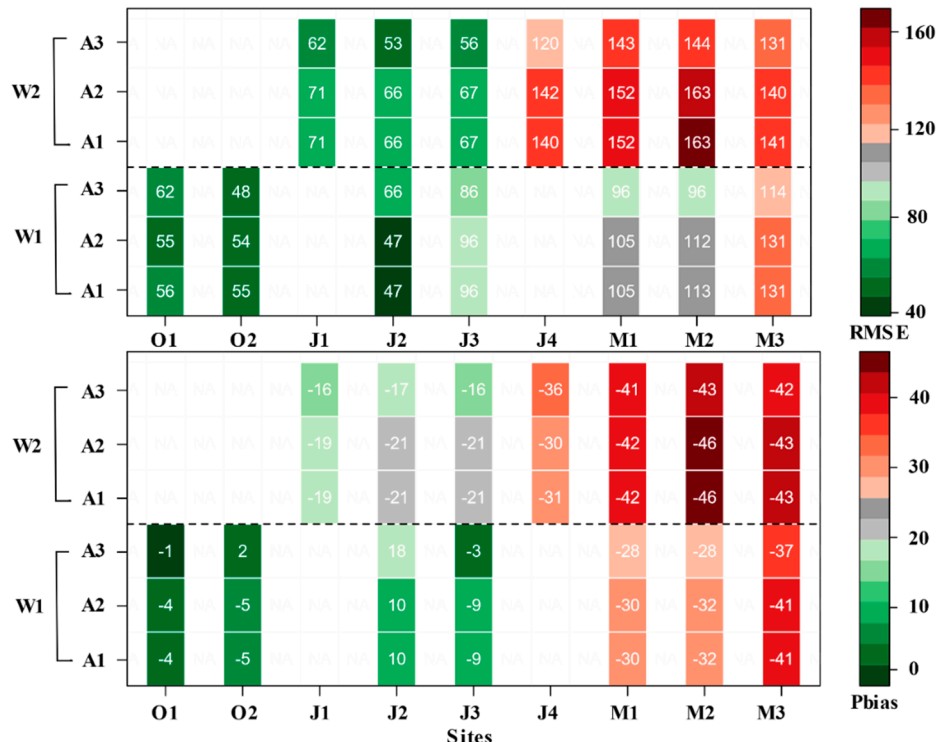

**Figure 4.** Performance of algorithms A1 to A3 in terms of the *RMSE* (mm w.e.) and *Pbias* (%). The colours of the bars denote absolute values.

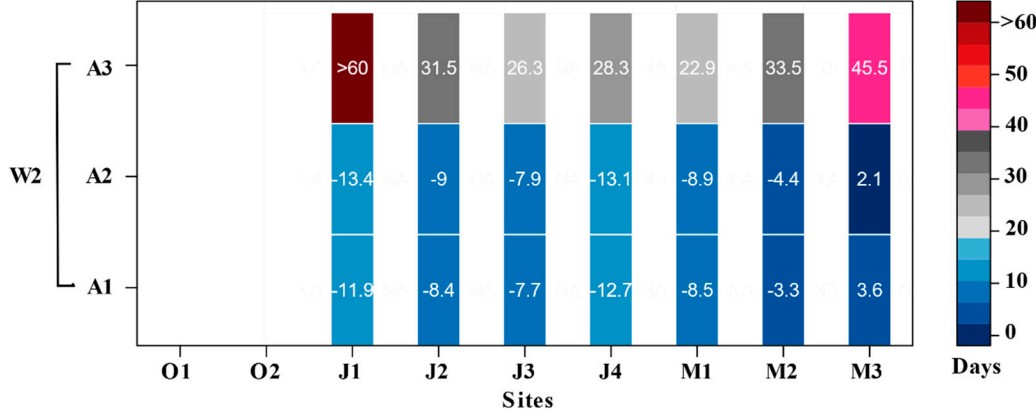

**Figure 5.** Observed versus simulated timing of the disappearance of the snowpack (days) for algorithms 1 to 3. Negative and positive values indicate anticipation and lag, respectively. The colours of the bar denote absolute values.

Overall, A3 was the best algorithm over the accumulation period, while A1 best simulated the disappearance of the snowpack. Therefore, A1 was selected as the base TI model to proceed with during the second testing phase (Figure 2).

### 3.2. Exploring Inputs into the Base Model During Phase II

Before proceeding to the second phase of the tests, some intermediate results were examined in order to determine which of the various inputs might be relevant. Despite their relative importance, inputs such as snow surface temperature and sublimation observations are mostly overlooked in TI models. Figure 6 presents the ratio of water vapour losses (mostly sublimation) to

bias-corrected incoming precipitation (top row) and the averaged air and snow surface temperatures at all sites (bottom row).

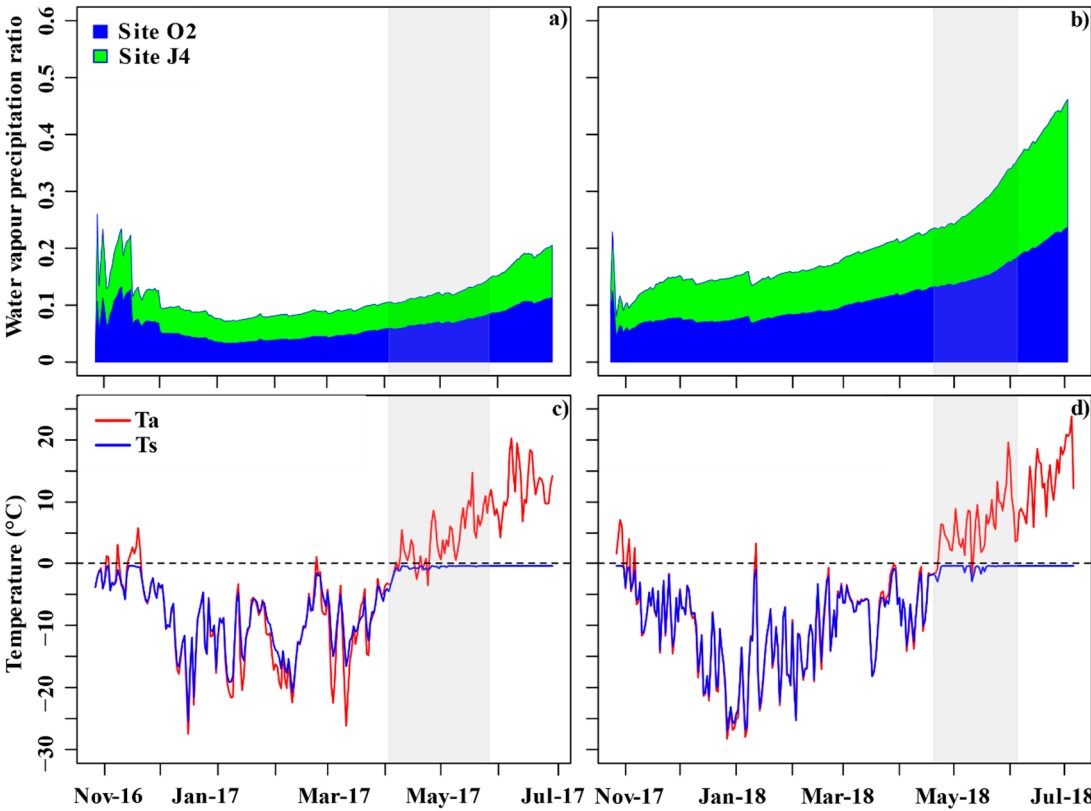

**Figure 6.** Importance of some relevant inputs: water vapour losses to precipitation ratio at daily time steps (**a**) for the first winter (W1) (**b**) for the second winter (W2) (**c**) Air ($T_a$) and snow surface ($T_s$) temperature (all sites averaged) at daily time steps (**c**) for the first winter (W1) and (**d**) air ($T_a$) for the second winter (W2). In the above plot, the shaded region represents the ablation period and to its left is the accumulation period.

During the accumulation period, at the juvenile forested site, the water vapour loss to precipitation ratio was on average 0.06 and 0.13 for the first and the second winters, respectively. For the same period, at the sapling forest site, with shorter vegetation, the mean ratio was 0.05 and 0.10 for the two winters. With the onset of spring, we observe an increase in the ratio for both winters (Figure 6a,b). However, there was a sharp incline for the second winter, mostly attributed to higher air temperatures (Figure 6).

By studying the time series of air and snow surface temperatures, we noticed the persistence of sub-freezing temperatures for most of the accumulation period (Figure 6c,d). On average, air temperatures were −3.3 and −3.8 °C for the first and the second winter, respectively. During the same periods, the average snow surface temperatures were −5.0 and −7.9 °C. During the ablation period, the second winter was 2.5 °C higher (6.1 °C) than the first (3.6 °C), while these temperatures were much more similar to one another during accumulation (i.e., −9.4 °C for W1 and −10.2 °C for W2).

### 3.3. Phase II: Including Additional Inputs/Processes in the Base TI Model

Various processes/inputs were inserted into the base TI model (A1) and the results are illustrated in Figure 7. Note that the insertion of certain inputs, such as sublimation, improved the performance of the base TI for both winters. Moreover, for the first winter, the inclusion of snow surface temperature ($T_s$) enhanced the SWE estimate for all studied sites (mean *Pbias* of 2.28% and *RMSE* of 7 mm w.e.).

In contrast, for the second winter, such insertions proved to be efficient only at juvenile forest sites (J1, J2 and J3). When snow interception was inlayed in the base TI model (for W1 and W2), it improved the model for mature forest sites (M1, M2 and M3) and for one juvenile site (J4) during W2. With the exception of sites O1 and J2 (for the first winter), including forest and radiation factors did not improve the base TI model. The addition of cold content improved the model at juvenile forest sites (J1, J2 and J3), but this improvement was not replicated at the other sites.

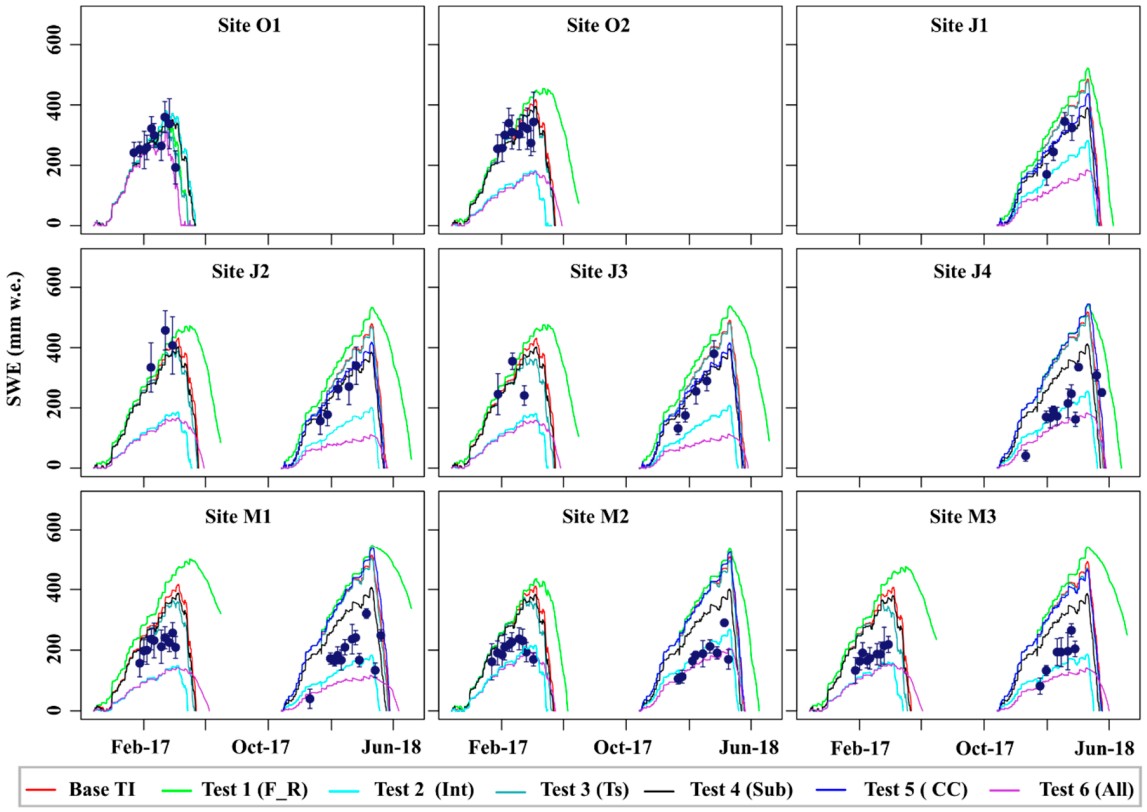

**Figure 7.** Hourly observed and simulated SWE for tests 1 to 6, using A1 as the base model. F_R indicates for forest and radiation, Int is snow interception, Ts is snow surface temperature, Sub is sublimation, and CC is cold content. The necessary inputs (snow depth and Ts) for test 5 were available for W2 only, limiting our test. Points illustrate the average SWE and the error bars denotes the standard deviation.

For the first (W1) and second winter (W2), adding the subcanopy radiation into the base TI model (test 1) resulted in reductions in performance of 9% and 3%, respectively (Figure 8). Although interception is a key process in forests, including this input in the base TI model did not result in a systematic improvement of performance across all sites. For instance, the addition of this input improved the base TI model at only two mature forest sites (M2 and M3) for both winters. At the other sites, it resulted in lapses in model performance by 120% and 85% for the first and the second winters, respectively.

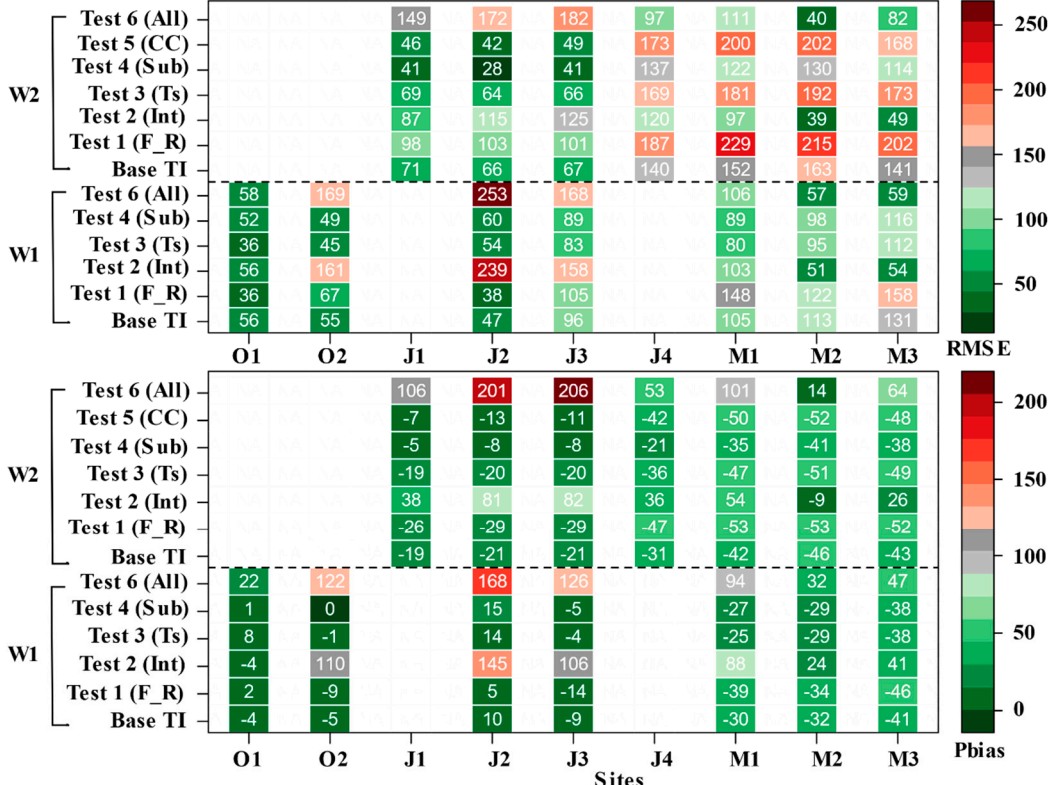

**Figure 8.** Model performance expressed in terms of the *RMSE* (mm w.e.) and *Pbias* (%) for all tests. *F_R* indicates forest and radiation, Int is snow interception, Ts is snow surface temperature, Sub is sublimation, and CC is cold content. The results are given separately for W1 and W2. The colours of the bar denote absolute values.

The inclusion of snow surface temperature as a threshold for melting (test 3) resulted in a slight modification of performance values. During W1, no clear improvements were detected, while, for W2, there was some small improvements at the juvenile forested sites (J1, J2 and J3) (Figure 8).

Unlike other inputs, the inclusion of sublimation (test 4) resulted in general performance improvement for both winters (i.e., 0.6% for W1 and 10% for W2). The addition of cold content (test 5) also proved beneficial. Although test 5 could not be conducted during W1, we noticed a clear improvement in the model performance at juvenile sites J1 to J3 for W2. Finally, the combination of all the additional inputs/processes (test 6) resulted in reduced performance values for all the tested sites (Figure 8).

## 4. Discussion

In the first phase of our study, three simple but common TI algorithms were explored for use in a small forested catchment area. All algorithms failed to simulate the spatiotemporal variability of SWE at most experimental sites (Figures 3 and 4). These simple TI algorithms yielded reasonable estimates for open sites with shorter vegetation (O1 and O2) but not at forested sites (M1, M2 and M3). Several previous studies reported similar limitations and proposed modifications to TI algorithms that included supplemental variables or processes such as solar radiation [3,29,31,65], variable melt factor [3,31,34–36], cold content [3,32,50], and other forest factors that lower the melting rate beneath the canopy [3]. A similar path was explored here, exploiting the best performing TI algorithm (A1) in a second phase by including these additional variables/processes. Algorithm A1 was judged to be more successful than the two others in capturing the evolution of SWE during both accumulation and ablation periods (Figures 3–5). As suggested by Ferguson [40], Kampf and Richer [41] and Magnusson et al. [66], a model with balanced parameterization generally leads to better estimates.

Reduced melt beneath the canopy was estimated using a factor based on the transmissivity of semi-permeable medium (the canopy), as detailed in Jost et al. [3] (Test 1). While the insertion of solar radiation provides energy [29], the forest canopy blocks incoming energy (lower transmissivity) and reduces snow accumulation [3]. In general, the results did not provide much support for this A1 modification, with the exception of the forest clearing site (O1) and one juvenile forest site (J2). Forest clearings are known to experience higher solar radiation penetration than canopy-covered locations [13]. The fact that the modification is only useful for an open site indicates that the canopy influence is not accurately represented.

Snow interception is known to affect accumulation and melt dynamics [17,51,60]. Based on an experiment conducted in the boreal forest of the Canadian Prairies, Pomeroy and Gray [67] reported that more than one-half of the total seasonal snowfall is intercepted [64]. In this study, algorithm A1 was modified for snow interception following the method used by Hedstrom and Pomeroy [60] (Test 2). Performance improvement results were varied at the mature forest sites (M2 and M3) but showcased deterioration at the juvenile forest sites. Leaf area index (LAI) [60] and canopy cover (Csc) [68] are typical inputs of snow interception components. Several studies have reported a weak relationship between the coefficient of variation (CV) of SWE and canopy features such as the LAI [51,69] and canopy density [70]. Figure 9 illustrates the snow depth measurements recorded by automatic snow depth sensors at sites M1, M2, and M3, along with some key forest characteristics. Both LAI and Csc at site M1 were higher than at the two other sites, which technically translates to more interception and reduced snow depth. Yet, we observed more snow depth at sites with higher LAI and canopy density. In contrast, there is a clear relationship between snow depth and tree height (TH) (Figure 9). Mazzotti et al. [71] studied the temporal relationship between snow depth and canopy variables (tree height, LAI, Csc, etc.) in the European Alps. They suggested including multiple canopy variables when studying snow accumulation and melt. Similar findings were reported by Parajuli et al. [51], who utilized statistical models to estimate the evolution of SWE based on the dataset that was also used in the present study. It is also worth noting that the interception component described in Hedstrom and Pomeroy [60] was developed in a mature forest (tree heights between 16 and 22 m) located in the much drier climate of Western Canada, while the present study watershed is exposed to a humid continental climate with variable stand structure. As LAI and canopy density are implemented to derive various snow-related processes such as snow interception [17,60,68] and canopy transmittivity [6,52,72], it is important to evaluate the measurement uncertainties associated with such variables to allow the resulting bias in different modelling studies.

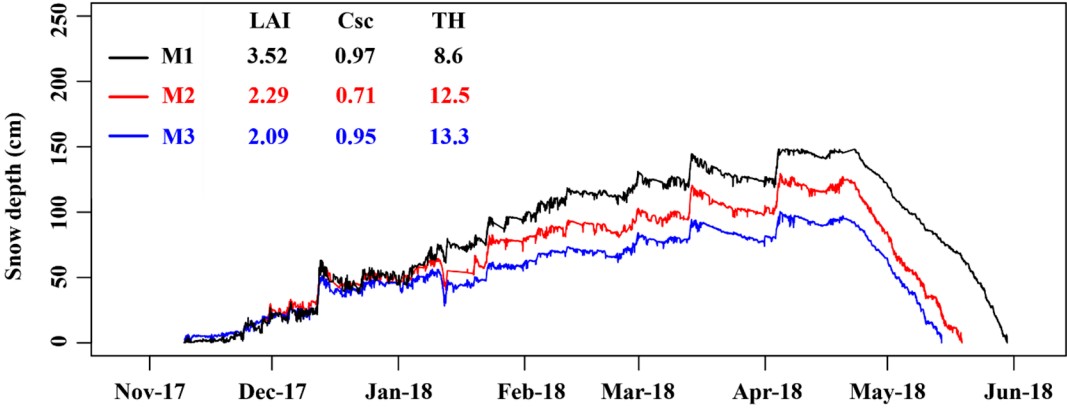

**Figure 9.** Snow depth recorded at the three mature sites (M1, M2 and M3) during winter 2. The corresponding leaf area index (LAI in $m^2\ m^{-2}$), canopy cover (Csc in fraction), and mean tree height (TH in m) are also provided.

Snow surface temperature ($T_s$, Test 3) improved the general performance of algorithm A1 for most of the studied sites, particularly in the first winter (Figures 7 and 8). For the second winter,

improvements were limited to the juvenile forest sites (J1, J2 and J3). The difference between air ($T_a$) and snow surface ($T_s$) temperatures is the primary driver of sensible heat flux [73], but, nonetheless, most TI implementations are limited to the use of constant threshold temperatures [3,29,31,41]. The results show that $T_s$ is a pertinent input, but its limited availability makes it an uncommon variable in TI models.

Sublimation (above and beneath canopy [62,74]) also improves performance values of algorithm A1 at most sites (Test 4). As pointed out by Molotch et al. [62], because sublimation observations are often limited to experimental studies, distinguishing processes above and beneath canopies are even more challenging, as it demands further logistics and greater cost. Nonetheless, Molotch et al. [62] successfully separated above and below canopy sublimation at Niwot Ridge Forest, Colorado, by utilizing eddy covariance devices. In this way, they were able to report sublimation values of 0.70 and 0.41 mm day$^{-1}$, respectively. Figure 6 illustrates the total sublimation to precipitation ratio for the two flux towers of the BEREV experimental watershed. Sublimation accounts for between 6 and 13% of the annual total precipitation at the juvenile site and 5 to 10% at the sapling site.

Adding the snowpack cold content did not provide any general improvement to algorithm A1 (Test 5), with the exception of sites J1, J2, and J3 (Figure 8). Deep snowpacks accumulate higher cold content compared to shallow ones [75]. The prevalence of deep snowpacks at juvenile sites (J1, J2 and J3) might explain the gains for these locations. Increased cold content results in delayed melt, which thereby extends snow persistence at the end of the season [76].

With the exception of sublimation and snow surface temperature, which were beneficial to algorithm A1, especially for the first winter, the other modifications were either neutral or detrimental. It is therefore not surprising that applying all the modifications together (Test 6) did not provide a good solution to the limitations of algorithm A1. There are three possible explanations for the substantial bias found in our simulation. First, our analysis was based on snow tube samples collected mostly during the accumulation period. As pointed out by Parajuli et al. [51], there is a certain level of measurement error associated with snow tube samples particularly under warm conditions, which brings further uncertainty during snowmelt. Second, we considered precipitation measured 4 km north from our studied site to be uniformly distributed, throughout the catchment, which might have contributed to the bias. Third, Parajuli et al. [51] reported the existence of spatiotemporal variability in SWE distribution, largely associated with differences in the stand structure. Regardless, using similar algorithms, Jost et al. [3] were successfully able to apply different variants of TI models to simulate the evolution of SWE in locations with a forest canopy in western Canada. They, however, optimized various empirical factors with respect to their algorithm. Unlike Jost et al. [3], we mostly relied on free parameters available in the literature (uncalibrated) without altering the base factors (Table 3) for any tested algorithm.

**Table 3.** Metrics for the best-performing TI model from Phase II (with sublimation) considering precipitation undercatch and precipitation phase comparison. BC = bias corrected, UC = uncorrected precipitation.

| Site | Metric | Precipitation Correction | | Precipitation Phase (Threshold Temperature) | | | |
|------|--------|------|------|------|------|------|------|
| | | BC | UC | 0 to 2 °C | −1 to 3 °C | 0 °C | 2 °C |
| O1 | *Pbias* (%) | 1 | 9 | 1 | 2 | 4 | −3 |
| | *RMSE* (mm w.e.) | 52 | 50 | 52 | 51 | 49 | 57 |
| M2 | *Pbias* (%) | 15 | 26 | 15 | 17 | 19 | 10 |
| | *RMSE* (mm w.e.) | 60 | 87 | 60 | 64 | 71 | 48 |

Our study has overcome the issue of wind-induced precipitation undercatch by utilizing manual DFIR observations to construct a robust (bias-corrected) precipitation dataset to feed TI models. Nonetheless, this type of data is rare, and uncertainty about the amount of precipitation remains an important issue that deserves our attention. The upper panel of Figure 10 reveals the impacts of

using bias-corrected vs. uncorrected precipitation as input to the best-performing TI model (including sublimation) at two of the study sites. The lack of bias correction on precipitation leads to a decrease in model performance, as *Pbias* values increase by 8% and 11% for sites O1 and M2, respectively (Table 3).

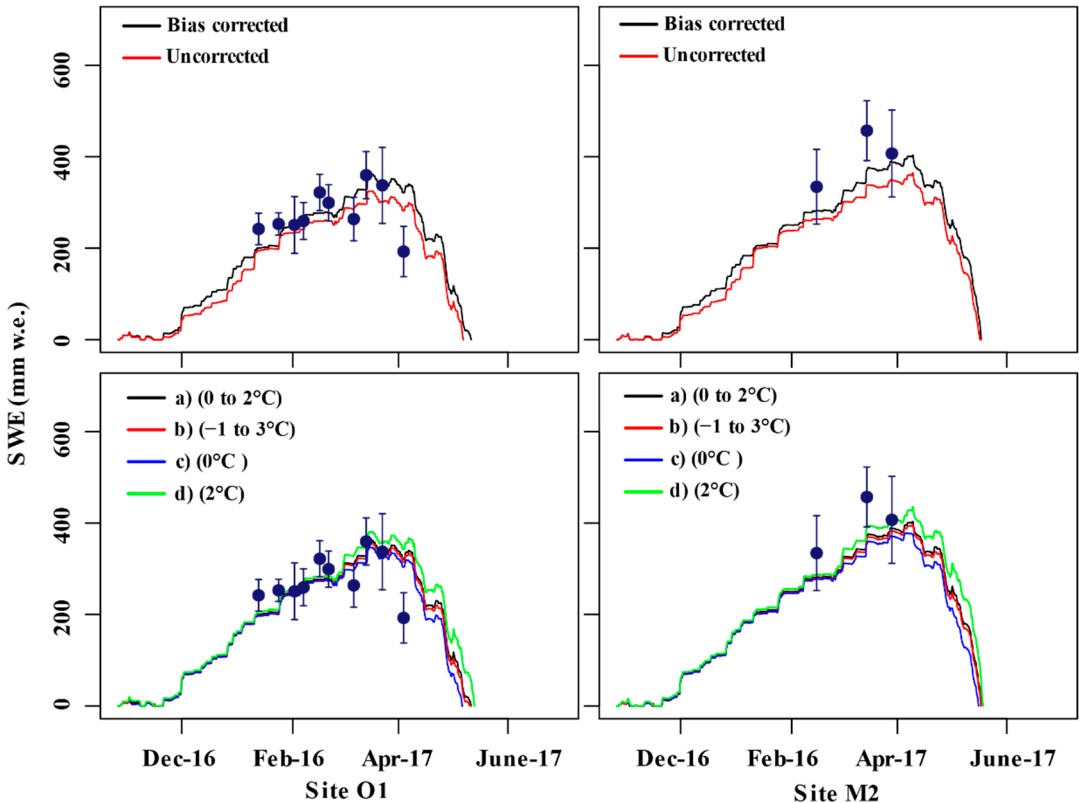

**Figure 10.** Effects of precipitation uncertainty (amount and phase) on modelled SWE at hourly time steps using the best-performing TI model from Phase II (with sublimation), at sites O1 (left) and M2 (right). *Top panel*: Modelled SWE using bias corrected vs. uncorrected precipitation data. *Bottom panel*: Impacts of various phase partitioning approaches. Four approaches are tested: a) the linear partitioning of rain and snow between air temperatures of 0 to 2 °C; b) the linear partitioning of rain and snow between air temperatures of −1 to 3 °C; c) snow and rain below and above 0 °C; d) snow and rain below and above 2 °C. Blue dots stand for the mean SWE, whereas the error bars denote the standard deviation.

In addition to the amount of precipitation, determining the phase (rain or snow) is critical. In our analysis, we used a linear distribution of snow/rain between 0 and 2 °C. The lower panel of Figure 10 illustrates the effects of various phase partitioning approaches using the same TI model and sites as the previous test. We notice some differences in the modelled SWE, as seen in Table 3. When using a fixed threshold of 0 °C, the *Pbias* values are 4% and 19% for site O1 and M2, respectively, whereas the performance is slightly better when using a fixed threshold of 2 °C (*Pbias* values of −3% and 10%). Using a linear distribution of snow/rain between −1 and 3 °C led to a similar output (*Pbias* of 2% for site O1 and 17% for site M2) than the range 0–2 °C we used in the study (*Pbias* of 1% and 15%). Obviously, the choice of a phase partitioning approach has an impact on the SWE modelling, but it is considerably small. Supported by an analysis combining 17.8 million observations across 29 site-years, Jennings et al. [22] carried out a study to identify the snow/rain temperature threshold across the Northern Hemisphere. The authors concluded such a threshold should be in the range −0.4 to 2.4 °C for 95% of the study sites. Hence, a linear partitioning of precipitation phase between 0 and 2 °C as implemented in this study appears as the best compromise.

Litt et al. [77] assessed a TI model on glacier ablation in the Nepalese Himalayas. They concluded that it is difficult to transpose these models to such environments and suggested exploring for

alternatives. Modelling snow processes within forests is equally challenging [78]. Variable stand structures induce irregular accumulation and melt patterns [79] adding complexity to snow processes. Implementing TI models in complex landscapes like glaciers or forested environments remains challenging, even when supplemental observations similar to those in our present study are available. We have seen many TI studies that argued that the simplicity and freedom of parameterization are major advantages when upscaling to catchment level [41,43,46,47]. These studies also attest to the strong correlation between air temperature and energy budget terms, such as sensible and latent heat flux and longwave radiation, providing the physical basis of TI models. However, implementing TI models and upscaling to the catchment level for hydrological simulations in complex landscapes such as forests may result in larger biases and erroneous simulations.

## 5. Conclusions

Located in a humid boreal forest (QC, Canada), the present study focused on understanding TI models and assessing their ability to simulate SWE time series in a complex environment. In the first step, three simple algorithms were implemented in order to identify the best possible base TI model for further testing. Although the more complex A3 algorithm performed better over the accumulation periods, the relatively simple algorithm A1 was selected because it was able to simulate the end of the snow season more accurately than algorithm A3. In the second step, algorithm A1 was modified to account for supplemental observations and processes. This resulted in few positive improvements. In most instances, modifications led to similar or reduced performance values. Snow interception improved the TI model only at mature forest sites. Incoming radiation was only beneficial at open sites. The cold content of the snowpack resulted in improved simulations at the juvenile forest sites. Only snow surface temperature and sublimation led to a general improvement of algorithm A1. The problem with the latter two modifications is that they resort to information that is rarely available. While it is quite inexpensive to measure snow surface temperature, sublimation demands high-frequency flux sensors that are expensive to deploy and operate. However, one may consider running a flux partitioning model such as the one proposed by Wang et al. [80] in lieu of flux towers. In the case where such tools are utilized, it may be a good idea to consider replacing the TI model altogether with a more physical energy balance modelling tool, particularly for complex environments such as forests.

**Author Contributions:** For this paper, A.P., D.F.N. and F.A. created the research/experimental design. A.P., O.S.S. and D.F.N. (occasionally) participated in the field trips to extract data. S.J. provided necessary sensors and dataloggers. A.P. performed data analyses and wrote the manuscript. D.F.N., F.A., O.S.S. and S.J. provided constructive feedbacks to improve this manuscript. All authors have read and agreed to the published version of the manuscript.

**Funding:** This work is part of the EVAP project, funded by the Natural Sciences and Engineering Research Council of Canada (NSERC), Ouranos (Consortium on regional climatology and adaptation to climate change), Hydro-Québec, Environment and Climate Change Canada and the Ministère de l'Environnement et de la Lutte aux Changements Climatiques through grant RDCPJ-477125-14.

**Acknowledgments:** Winter field measurements for this study were possible thanks to Professor André Desrochers, who kindly provided us access to snowmobiles. The authors are indebted to François Larochelle and Martine Lapointe for their initial support, with the design of the instruments. We would like to express deep gratitude to Annie-Claude Parent who managed logistics, participated in the field campaigns and supported our research from start to end. Thank you to Benjamin Bouchard and Médéric Girard, who both contributed to the winter field campaigns. Montmorency Forest staff, including Robert Côté and Charles Villeneuve, are acknowledged for their generous support and their immense help with logistics. Thanks also to Fabien Gaillard Blancard, Jonas Götte, Kelly Proteau, Bram Hadiwijaya, Pierre-Erik Isabelle, Judith Fournier, Alicia Talbot Lanciault, Georg Lackner, Carine Poncelet, Amandine Pierre, Guillaume Hazemann, Marco Alves, Adrien Pierre, Antoine Thiboult and the PEGEAUX members who also participated in the field campaign. The authors would like to acknowledge Professor Stephen Déry and two anonymous reviewers for enhancing our manuscript by providing constructive feedback.

**Conflicts of Interest:** The authors declare no conflict of interest.

**Data Availability:** The data that supports the findings of this study are available from the corresponding author upon reasonable request.

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
