# Peer review of "Does Data Availability Constrain Temperature-Index Snow Models? A Case Study in a Humid Boreal Forest"

_water, doi:10.3390/w12082284_

Round 1
Reviewer 1 Report
This study investigated impacts of data availability on the accuracy of temperature index snow model at a study area in Canada. The topic of this study seems to be interesting. However, the methodology is not described well. Currently, It is difficult to properly evaluate this study.
Specific comments:
The methodology is not described well in this manuscript. It is difficult to understand the purpose of each comparison. Meanwhile, there are various errors in modeling, and the effects of the errors are very complicated. Removing a kind of error in modeling does not always increase the accuracy of the modeling. For instance, Test 6 did not show the best results in this study. Currently, it is difficult to understand the importance of the comparisons in this study.
Please avoid using a one-sentence paragraph.
Please use the same font for the variables between in the equation and in sentences.
L.56: “Besides, difficulties in the spatial interpolation of variables, such as wind speed and incoming radiative fluxes, hinder their applicability at larger scales” Nowadays, atmospheric reanalysis data or its downscaled data can be used as input to a snow model.
L.141-142: Please briefly describe the topography of the study area.
L.146-148 : Please add a reference for this description.
L.152-154 : “Turbulent fluxes were analyzed over 30-min spans and averaged to be on an hourly scale like the rest of the model inputs.” This sentence is not clear.
L.152-164 : “Based on numerous snow pit surveys, the snowpack temperature during that period typically ranged between −0.2C to −0.7C, thus existing gaps were replaced by a constant temperature of −0.45 C.” Please add references.
L. 165-166: Is the difference between the actual snow surface temperature and the recorded data closest to the snow surface negligible for the simulations in this study?
L.172 : Please define “the base TI model” before use.
L.173-175 : Please explain it in a little bit more detail here.
L.178-180 : Please define Tair. Also, what is the threshold temperature?
L.185 : Why is the unit of the end of daylight “h”?
L.201 : Please explicitly describe the difference between Equation 1 and 5. Is Equation 6 not used for Equation 1?
L. 225-226: This description is not clear.
L.228-231 : Please add more detailed explanation. What are compared? What is the purpose?
L.229 : Does “the model” in this sentence the TI model?
L.234: rho_s -> rho_sa?
L.235-236 : “As snow density undergoes metamorphism with time, the formulation utilized in Equation 9 fails to come up with accurate estimate.” Please add a reference. If none, please explain it in more detail.
L.239 : I cannot find Ta in Equations 10 and 11. Is Ta related to Equation 10?
Figure 3: It is difficult to see the red line in the plots.
Figure 3: Why does the observation have the standard deviation?
L.266-270 : The observation seems not to be
a single value at each time. How were RMSE and Pbias calculated? Was the mean value used?
L.280-289 : What are differences between this paragraph and the previous paragraph?
L.284-284 : “automatic snow depth measurements and field observations” Please explain these data in the Data inputs section.
L.297-299 : It seems that A1 is almost similar to A2.
Figure 6: Please explain the panel c and d in the caption.
Author Response
The authors would like to acknowledge the Editor and the two anonymous reviewers for providing valuable comments on our work. We have made nearly all of the requested revisions and have provided proper justification otherwise. We hope that the reviewers will find this new version satisfactory. Note that we sent the paper to a professional English editing service as recommended by the Editor, hence one can observe large number of edits.

Reviewer 2 Report
In Does data availability constrain temperature-index snow model? A case study in the humid boreal forest, the authors modify three versions of the common temperature index snow model to investigate if incorporating additional data-informed mechanisms can improve snowmelt modeling. The paper was fairly clear (some minor comments below) and touched on some of the key challenges of snowmelt modeling, such as diverse canopy structures, differences in different climates, and lack of widespread data availability. Ultimately, the authors concluded that temperature index models are not typically adequate for snow modeling in heterogeneous watersheds.
One challenge of snow modeling that was not thoroughly addressed was the uncertainty of precipitation in both amount and partitioning. Although the author’s bias corrected their precipitation results (partially addressing one of the largest sources of uncertainty in snow modeling per Raleigh et al 2015), literature suggests that precipitation partitioning between rain and snow in some climates is not well captured by a simple temperature threshold (e.g. Wayand et al 2016). Did the authors consider the error associated with their assumption in rain/snow partitioning or if their relatively simple precipitation bias correction process was adequate? Additional discussion on these topics would improve the completeness of the paper.
Minor comments
- Ln 35: “dynamics…are more complex”
- Ln 36: “Over the years, there have been substantial efforts”
- Ln 89-90: “The availability of such a dataset allows to articulate the following research question.” Incomplete sentence.
- Ln 323: The paragraph starting here should be checked for consistency of verb tense.
- While I like the visualization of the error statistics eg (Figures 4, 5, 8), the plots could be cleaned up a bit to remove excess white space.
- Figure 8 – Using the same color scheme as that in Figure 4 would improve consistency and prevent the appearance of neutral change in the middle of the color bar.
References
Raleigh, M. S., J. D.Lundquist, and M. P.Clark, 2015: Exploring the impact of forcing error characteristics on physically based snow simulations within a global sensitivity analysis framework. Hydrol. Earth Syst. Sci., 19, 3153–3179, doi:https://doi.org/10.5194/hess-19-3153-2015.
Wayand, N. E., J.Stimberis, J. P.Zagrodnik, C. F.Mass, and J. D.Lundquist, 2016a: Improving simulations of precipitation phase and snowpack at a site subject to cold air intrusions: Snoqualmie Pass, WA. J. Geophys. Res. Atmos., 121, 9929–9942, doi:https://doi.org/10.1002/2016JD025387.
Author Response

(The authors gave the same response as above.)
